# Disturbed Interstitial Pregnancy: A First Case of Successful Treatment Using a Mini-Laparoscopic Approach

**DOI:** 10.3390/medicina55050215

**Published:** 2019-05-27

**Authors:** Latchesar Tantchev, Andrey Kotzev, Angel Yordanov

**Affiliations:** 1Obstetrics and Gynecology Clinic, Acibadem City Clinic Hospital “Tokuda”, PC 1000 Sofia City, Bulgaria; 2Clinic of Gastroenterology, University Hospital for Active Treatment “Alexandrovska”, PC 1000 Sofia City, Bulgaria; dr_andrey_kotzev@abv.bg; 3Clinic of Oncological Gynecology, University General Hospital for Active Treatment “Dr. G. Stranski”, PC 5800 Pleven City, Bulgaria; angel.jordanov@gmail.com

**Keywords:** mini-laparoscopy, minimally invasive surgery, interstitial pregnancy, ectopic pregnancy of tubal stump

## Abstract

Interstitial ectopic pregnancy (EP) is a life-threatening condition due to the risk of massive hemorrhage in the event of its disturbance. We present the case of a 27-year-old patient who was admitted with massive hemoperitoneum, caused by the rupture of an interstitial pregnancy in the area of the fallopian tube stump, which had been removed after a previous ectopic pregnancy. The condition was overcome using a mini-laparoscopic approach (2.6 mm, 30° optics), with one 3 mm port for micro-laparoscopic instruments and one 10 mm port. Such an approach has not yet been reported in the available literature, among the casuistically reported cases of pregnancy in the tubal stump. We consider that the technique is safe, completely in the interest of the patient, applicable by an experienced team, and in agreement with modern trends regarding the minimization of operative access.

## 1. Introduction

Ectopic pregnancy (EP) presents a serious risk to the life of patients in reproductive age. The condition is connected with conception later in life, increased frequency of pelvic inflammatory disease, application of assisted reproductive technologies (ART), and tubal or pelvic surgery [1]. The incidence of ectopic pregnancy is 1.3–2% of all pregnancies [2], while 2.5% develop in the interstitial part of the uterine tube [1,3].

A particular case of interstitial pregnancy is the nidation in the stump of a removed uterine tube, with an incidence of only 0.4% [4]. A certain number of authors use the terms interstitial and cornual pregnancy (rudimentary horn pregnancy) as synonyms. According to Botros et al., pregnancy is defined as cornual when it occurs in a rudimentary horn, unicornual uterus, bicornual uterus, or uterus didelphys [5]. The terminological difference is important, because the therapeutic measures adopted in one or the other type of ectopic pregnancy vary. The basic approaches in EP therapy are conservative ones, accompanied with surgical treatment. Surgical intervention becomes necessary when the pregnancy is disturbed and hemoperitoneum is present. Due to the anatomical specifics of the uterine horn’s blood supply, especially during pregnancy, rupture in that area would bring about life-threatening hemorrhage. That is why some authors recommend performing a laparotomy with cornual resection or a hysterectomy [5]. The laparoscopic access, when EP is present, has been recently accepted as a “gold standard” [6]. The pursuit of an even faster recovery, the reduction of operative trauma, and a better cosmetic effect has resulted in the introduction of mini-laparoscopic approaches in practice. No precise definition of the term “mini-laparoscopy” is found in the scientific literature, although various authors define it as an intervention performed by instruments with a diameter of 2 to 5 mm, except for the diameter of the umbilical opening [7].

## 2. Case Report

This case concerns a 27-year-old female patient with two previous pregnancies—a tubal pregnancy, which ended in a laparoscopic left-sided salpingectomy, and an intrauterine pregnancy, which ended in the parturition of a full-term newborn after cesarean section. The patient was admitted as an emergency case—hemodynamically stable, with a severe, piercing pain in the lesser pelvis, positive Blumberg’s sign, and echographic data of hemoperitoneum. The patient reported a positive urine pregnancy test, according to the term of amenorrhea in 7.2 gestational weeks, while hyperplastic endometrium of 14 mm was visualized by the vaginal echography performed without presence of an intrauterine pregnancy. The laboratory parameters at the time of admission were as follows: hemoglobin—127 g/L, hematocrit—0.374 l/L, β-human chorionic gonadotropin (β-HCG)—9957.96 mIU/mL.

Emergency mini-laparoscopy was performed in view of the imaging and clinical data of disturbed EP and hemoperitoneum. Massive hemoperitoneum was found with the presence of sanguineous coagulums in the lesser pelvis, lateral paracolic gutters, and domes of the diaphragm.

When performing the intervention, we used 2.6 mm, 30° optics (LIL-33-30, Microlap, Conmed, Utica, NY, USA), with one 3 mm port (Microlap, Conmed, Utica, NY, USA) for micro-laparoscopic instruments and one 10 mm port for evacuation of decidual portions and sanguineous coagulums, and insertion of a needle and 15 mm, 2-0 V-Loc™ suture. A set of instruments was used for mini-laparoscopy (Microlap, Conmed, Utica, NY, USA). The operative access was realized with a Veress needle in the base of the umbilical ring.

Active arterial bleeding from a rupture of interstitial pregnancy was found in the area of the left uterine horn, at the site of a previous salpingectomy. Coagulation was performed and hemostatic suture was placed (Figure 1).

Decidual portions and sanguineous coagulums were evacuated, and lavage and drainage of abdominal cavity were performed. A cyst of yellow body was visualized in the left ovary. No pathological changes were found in the right uterine tube or uterus. The duration of the operative intervention was 65 min, and there were no complications during its course. Hemotransfusion of one unit of red blood cell concentrate was realized in the early postoperative period; it was implemented due to the measurement of a reduction in the preoperative hemoglobin with 45 units at post-surgery hour 4. The abdominal drain was patent, but there was no demonstration of active bleeding. The patient was discharged on post-surgery day 3 in a good general condition with the following laboratory parameters: hemoglobin—86 g/L, hematocrit—0.265 l/L, β-HCG—2925 mIU/mL. 

Positive results of β-HCG were not found in a blood sample one month later. The surgical wounds healed by first intention.

Four months after the intervention was performed, an intrauterine pregnancy was found, with a cyst of yellow body in the right ovary; at present, it is following a course with no complications. 

The patient signed the consent for publication. The study was conducted in accordance with the Declaration of Helsinki, and the protocol was approved by the Ethics Committee of Acibadem City Clinic Hospital “Tokuda” under 22.03.2019/№22.

## 3. Discussion

The interstitial part of the uterine tube is localized in the musculature of the uterine wall, with a diameter of 0.7 mm and a length of 1–2 cm, and with a slightly folded course. When salpingectomy is performed, the interstitial part of the uterine tube is most often not removed. In the case described by us, the remainder of the uterine tube was not visualized due to a previous rupture. There are two basic supposed mechanisms by which a pregnancy in the interstitial part of a removed uterine tube can occur: the fertile egg can pass through an opening in the uterine stump and become implanted; according to the other theory, the zygote passes through the intact uterine tube, and thereafter implants itself in the interstitial part of the removed one. In view of the presence of yellow body in the left ovary, the first mechanism for the occurrence of the EP is more probable as far as our case is concerned. Transperitoneal migration of the egg from the left ovary through the right uterine tube is also theoretically possible with subsequent nidation in the interstitial part of the left uterine tube. The exact mechanism cannot be adequately established in this particular situation.

The only method for treatment, as far as the presented case is concerned, was the operative one due to data of present hemoperitoneum and active bleeding. An early diagnosis could have allowed a conservative approach by means of the systematic or local application of methotrexate [8,9]. We chose a mini-laparoscopic access due to the hemodynamical stability of the patient, and the presence of a sufficiently experienced laparoscopic team. The accurate diagnosis was made in the course of the operation, because when the first echographic examination was performed, the gestational sac was not visualized. The trend for adopting a laparoscopic approach, even in the cases of disturbed ectopic pregnancy and hemoperitoneum of more than 600 mL, has been established upon examination in chronological order of the cases presented in the available literature [4,6].

We used a mini-laparoscopic approach in the pursuit of minimizing surgical trauma, as well as ensuring concomitant benefits for the patient, such as the current instrument set and a well-trained surgical team. Mini-laparoscopic surgery in gynecology has a history of almost 30 years. Dorsey and Tabb reported on diagnostic interventions and myomectomies, performed by them with an optic laser using 3 mm incisions for the first time in 1991 [10]. The technique was used for a hysterectomy performed by A. Wattiez et al. in 1999 [11]. The interventions for treatment and determination of the stage of cancer of the endometrium (2009) [12], as well as of cancer of the uterine cervix (2011) [7,13,14], were also subsequently minimized. The method is applied for sacrocolpopexy with utero-vaginal and apical prolapse [15]. In a scientific report from 2015, the authors successfully demonstrated a mini-laparoscopic approach adopted for the removal of enlarged myomatous uteri at more than 16 gestational weeks [16]. Among the literature reference studies examined so far, we have found no application of that access—either in the presence of an interstitial pregnancy or in an emergency setting. We consider that the minimization of access did not bring about an extension of surgery time, despite the technical difficulties. Due to the presence of major amounts of blood and coagulums in the peritoneal cavity, illumination of the field was reduced compared to the classical laparoscopy with the use of 10 mm optics. Nevertheless, the quality of the image allowed the safe performance of the operative intervention. We encountered some difficulties due to the consistent insertion of the instruments in the 3 mm trocars and the insufficiently high flow of gas with bigger losses or the necessity of evacuating more smoke. That peculiarity requires a limitation to the minimum of gas losses, as well as the necessity to remove excessive coagulation, causing the presence of smoke, that is, it requires a team showing good collaboration and precise work that did not allow unintentional bleeding. In this context, the surgeon must have sufficient experience with conventional laparoscopy in order to avoid situations causing significant difficulties for the intervention’s outcome.

The patient tolerated the surgical intervention well and was discharged on post-surgery day 3 without complications upon reduction of the β-HCG level. 

## 4. Conclusions

A minimally invasive approach is preferred and may be safely applied, in the case of a disturbed interstitial pregnancy, by an experienced team and when the conversion to a laparotomy is assured. Mini-laparoscopy is an operative technique comparable to conventional laparoscopy in its clinical effectiveness. Access is completely in the interest of the patient, and despite the technical difficulties, we consider that it is applicable in interventions of major complexity, and that it will establish itself in the future as the preferred approach in accordance with the aspiration to reduce operative trauma.

## Figures and Tables

**Figure 1 medicina-55-00215-f001:**
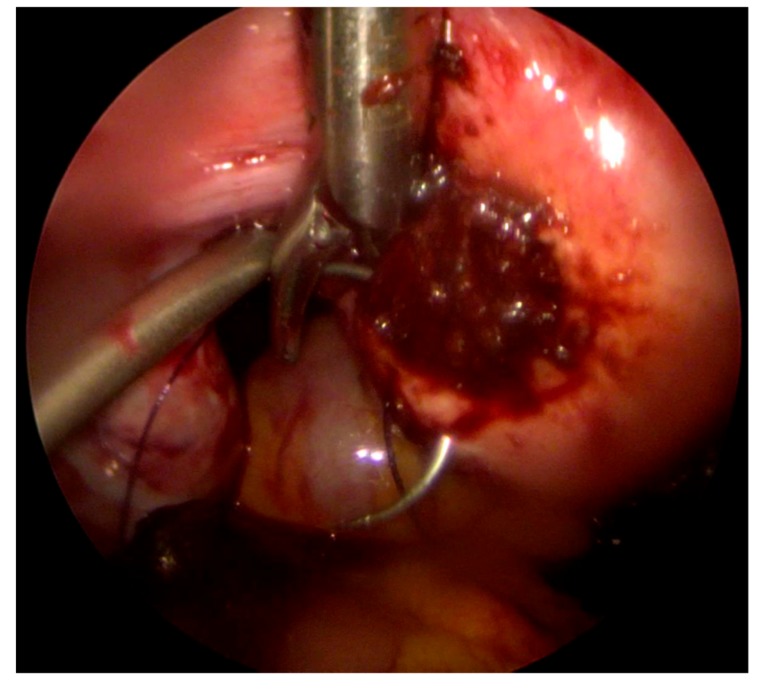
Mini-laparoscopic suture of the left uterine horn.

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
