# Peer review of "Disturbed Interstitial Pregnancy: A First Case of Successful Treatment Using a Mini-Laparoscopic Approach"

_medicina, 2019, doi:10.3390/medicina55050215_

Round 1

Reviewer 1 Report

The Authors reported a case of Interstitial pregnancy treated by mini-laparoscopy

Overall the case report is interesting and well written.

Title: adequate

Abstract: adequate

Key words: correct, minimally invasive surgery may be added

Introduction: Few words about mini laparoscopic approach in surgery may be added

Case report: well documented

The paragraph “Active arterial bleeding from a rupture of interstitial pregnancy was found in the area of left uterine 73 horn, at the site of a previous salpingectomy. Coagulation was performed and hemostatic suture 74 was placed. (Fig 1)” can be moved after  “umbilical ring”.

Discussion: few words on Mini lap in gynecological field may be spent. I think that the Authors should stress this point: a significant experience in laparoscopic surgery is basic before embarking in this approach.

Figures: A picture of Post operative cosmetic results would be perfect.

Author Response

Thank you for the revision, we tried to make the manuscript better, by the following corrections: 

Introduction - a short definition was added

Case reprt - the paragraph was moved

Discussion - a short descroption of the application of mini-laparoscopy in gynecology was added, which changed the bibliography, and further explanation of the need od well trained laparoscopic team was given.

Unfortunately we don't have a picture for the cosmetic result in the concrete case.

P.S. A very recent intrauterine pregnancy was diagnosed, which was added to the text as a fact.

Reviewer 2 Report

The reported case describes the use of minilaproscopic approach to treat an interstitial pregnancy occurring on the tubal stump in a women who underwent  tubal excision for a previous ectopic pregnancy. Even though this approach can be included in the potentiality of minilaproscopy, the case described is actually rare and could be of interest for the reader.

The article is well written and the contnt is clear.

More information should be given on the operative time needed to perform the surgery in this case.

Authors could discuss the choice to perform a minilaproscopy in an emergency setting. 

Author Response

Thank you for the revision, we tried to make the manuscript better, by the following corrections: 

The missing operative time was added. We chose a mini-laparoscopic access due to the hemodinamical stability of the patient, and the presence of expirienced laparoscopic team.

We did not find in the literature application of the access in emergency setting 

P.S. A very recent intrauterine pregnancy was diagnosed, which was added to the text as a fact.